# Review of the Forensic Applicability of Biostatistical Methods for Inferring Ancestry from Autosomal Genetic Markers

**DOI:** 10.3390/genes13010141

**Published:** 2022-01-14

**Authors:** Torben Tvedebrink

**Affiliations:** 1Department of Mathematical Sciences, Aalborg University, DK-9220 Aalborg, Denmark; tvede@math.aau.dk; 2Section of Forensic Genetics, Department of Forensic Medicine, Faculty of Health and Medical Sciences, University of Copenhagen, DK-1165 Copenhagen, Denmark

**Keywords:** ancestry, biostatistics, clustering, classification, distance based, likehood, hypothesis tests

## Abstract

The inference of ancestry has become a part of the services many forensic genetic laboratories provide. Interest in ancestry may be to provide investigative leads or identify the region of origin in cases of unidentified missing persons. There exist many biostatistical methods developed for the study of population structure in the area of population genetics. However, the challenges and questions are slightly different in the context of forensic genetics, where the origin of a specific sample is of interest compared to the understanding of population histories and genealogies. In this paper, the methodologies for modelling population admixture and inferring ancestral populations are reviewed with a focus on their strengths and weaknesses in relation to ancestry inference in the forensic context.

## 1. Ancestry Informative Markers

The increased availability of whole human genome sequences in the public domain is an invaluable data resource in many genomics, biomedical, and anthropological research areas. In particular, the data repositories focusing on the genomic diversity among human populations (e.g., HapMap Project [1], 1000 Genomes Project [2], Simons Genome Diversity Project [3], and more recently curated in the Genome Aggregation Database Project, gnomAD [4]) have contributed to the understanding of human evolution, migration histories, waves, and patterns.

Typically, genetic samples from populations with different geographical locations, cultural backgrounds, tribal memberships, and linguistic groups constitute the data used to identify genomic differences among the derived populations. These genomic differences are the result of a mixture of causes: mutations, recombination, genetic drift, selection, and migration [5]. The variations in the human genome can be observed both in tandemly repeated DNA sequences (e.g., short tandem repeats, STRs) and in single-nucleotide polymorphisms (SNPs) but are also manifested in sequence variations in lineage markers (e.g., Y-chromosome and mitochondrial DNA) and structural variations (e.g., copy-number variation) [6].

The geography of human populations (ancestry) and genetic polymorphisms are closely related [7,8,9,10,11,12]. Hence, identifying ancestry informative markers (AIMs) can be accomplished by analysing the publicly available genome sequences. Several measures of informativeness has been derived in order to rank candidate markers [13,14].

In forensic genetics, STRs are presently the standard markers used for identification and relationship testing. Furthermore, variation in allele frequencies may be used to infer an individual’s ancestry [15,16]. Other types of AIMs are microhaplotypes [17], which are groups of closely located SNPs (often positioned within 500 bp), and insertion–deletion polymorphisms (indels) [18,19]. However, SNPs are more commonly used for ancestry analysis [20,21], where several commercial assays have been developed for ancestry investigations. The selection of SNPs to be included in commercial kits depends both on their biochemical properties (e.g., allele balances and primer site location) and their informativeness regarding specific continental and regional populations. Hence, the companies focused on specific population structures (e.g., intracontinental differences) may result in poor resolution in other regions and among populations of close proximity. Consequently, all results obtained from the genotyping and subsequent analysis are conditioned on the specific SNPs included in the analysis. This is important to bear in mind when interpreting the results of an analysis, as general patterns and expectations to population variability may not be contained on a narrowly selected set of markers.

## 2. Biostatistical Methods

The study of population structure from genomic data has a long history within the genetic and statistical literature. The increasing complexity and availability of data from ancestry informative markers on multiple populations have required the development of appropriate methodologies to model and capture the subtle genetic structures in the data: from the first studies of a few hundred markers and individuals to more recent studies with thousands of individuals genotyped on highly dense assays or from whole genome sequences.

This progress, driven by biotechnological advances, has been challenging not only from a methodological point of view but increasingly so from a computational perspective. Where independence between the markers was ensured by their genetic distance, the closer proximity of genetic markers results in statistical dependence (e.g., due to linkage disequilibrium, LD). This increases the complexity of the analysis since such dependencies must be adequately modelled and their dependence structure learned from the data. Some methods incorporate the structure into their modelling framework, either by identifying stretches supposedly inherited as a complete block (e.g., [22]) or attempts to learn the association structure using graphical models [23]. Others operate by removing the dependence among markers by pruning or thinning the data based on LD (e.g., [24]) or using the residual signal after successively regressing the markers on each other (e.g., [25]).

The most widely used methodologies for analysing genetic structure can broadly be divided into four different groups: principal components analysis, model-based clustering, classification and likelihood-based, and hypothesis test-based methods. In addition, tree-based and distance/dissimilarity clustering methods are used but not included in this review. Most of the development in the study of population structures is driven by interests in medical genetics or population genetics, where controlling for population stratification may be vital for some applications or where the study of population dynamics on its own is the main purpose. Hence, few methods have been developed with the applications of forensic genetics in mind. In the subsequent sections, the various methodologies are discussed with references to key publications in their domain.

To demonstrate and visualise the results from the various methods, data from [26] are used in the following sections. The data consist of samples from six reference metapopulations (comprised by a total of $n_{\mathrm{ref}}=3453$ samples) and $n_{\mathrm{test}}=517$ test samples all genotyped on the 164 ancestry informative SNPs included in the Precision ID Ancestry Panel (excluding marker rs10954737 due to a high degree of missingness). The reference samples are known to originate from the indicated metapopulation, are of high sequencing quality, and used to estimate the metapopulation’s allele frequencies. The 517 test samples were harvested online, where their alleged metapopulation of origin was derived from the sampling location and other available meta information [26]. Both the reference and test samples originated from six regional areas, where the regional specific $n_{\mathrm{ref}}/n_{\mathrm{test}}$ counts are: Sub-Saharan Africa (668/37), North Africa (283/49), Europe (1014/173), Middle East (377/52), South/Central Asia (489/85), and East Asia (622/121).

### 2.1. Principal Components Analysis

Principal components analysis (PCA) has a long history of application in the study of population structure. The usage of PCA in the analysis of population structure was pioneered by [27], as an efficient way to capture the underlying structure for visualisation purposes. PCA is able to capture continuous admixture between populations, implying that admixed populations typically falls on the *lines* between its *parental* populations. More recent PCA-based methods for inference include SMARTPCA [25] and EIGENSTRAT [28]. These methodologies provide further insight as to how many PCs are required to capture detectable population structures by the use of hypothesis testing on the magnitude of the PCA’s eigenvalues. For the reference samples (cf. above) from [26], there are four significant PCs according to EIGENSTRAT. The first three PCs are plotted in Figure 1 (plots including the fourth PC do not visually separate the metapopulations). The often detected *triangle* spanned by Sub-Saharan Africa, Europe, and East Asia is cleary visible in the plot of PC1 and PC2.

Besides being an efficient tool for visualising high-dimensional (genetic) data, PCA also yields the best linear low-dimensional approximation of the data in terms of Euclidean distance. This, together with the orthogonality of the principal components (PCs), implies that population substructures can be detected by the discriminant analysis of the PCs, DAPC [29]. Primarily relying on linear algebra, DAPC is computationally much faster than STRUCTURE and gives comparable results. However, several papers discuss some of the known pitfalls when using PCA [30,31,32,33]. The most important issue to be aware of is PCA’s sensitivity to the sampling of individuals and populations. An unbalanced sampling of some populations forces the PCs to account for the variation caused by the majority groups. Since most models and methods try to minimise a *loss function*, this implies that in the estimation of the unknown quantities (e.g., model parameters, here, the PCs) emphasis is given to the majority groups in the data [30,31]. The loss function is typically proportional to the deviation between the observed data and the model’s predictions (or expectation), e.g., measured by the model’s likelihood or squared distance of the residuals. For PCA, the loss function is inversely proportional to the fraction of explained variance of the first PCs, with DAPC’s loss also depending on the intracluster variances as a function of the numbers of cluster *K* (measured by a Bayesian Information Criterion, BIC). Another property shared by several methods and statistical models is that they provide a *compression* of the data, in the sense that the methods are trying to retrieve as much information from the data as possible and discard a nonsignal as noise. Taken a bit further, one may interpret the model output and parameter estimates as *data summaries* (some more informative, advanced, and interpretable than others). Consequently, some of these summaries (e.g., the PCs and STRUCTURE as discussed next) are similar or even identical for different population genealogies [31].

### 2.2. Model-Based Clustering Methods

The seminal STRUCTURE paper [34] introduced a population genetics methodology based on a statistical finite mixture model. Using a Bayesian approach, the posterior probabilities for cluster memberships and cluster-specific allele frequencies are estimated using an MCMC-algorithm (MCMC: Markov chain Monte Carlo). STRUCTURE has been used in numerous studies and has, according to Google Scholar, more than 30,000 citations, which indicates its enormous influence in the field of population genetics [35]. Forensic applications were mentioned as one of motivations for STRUCTURE ([34], p. 945) in order to identify cryptic population structures and assess their influence on the detection of immigrants and calculation of match probabilities [36,37,38]. In essence, the initial STRUCTURE model is a simple Hardy–Weinberg model with the relaxation that subpopulations have different allele frequencies and individuals may inherit alleles from several subpopulations, i.e., being admixed [35]. Following the initial publication, several modifications both on the population genetic model (allowing for, e.g., linked markers and null alleles) [22,39] and computational aspects (faster and more efficient algorithmic schemes) [40,41,42] have been suggested. See [35,43] for further references and remarks.

Despite the many successful applications of STRUCTURE (including variants and similar methodologies, e.g., FRAPPE [40] and ADMIXTURE [41]), to population genetic data and research questions, no model is better or more general than its underlying assumptions. Similar stories apply to simple linear regression models, where the assessment and evaluation of the residuals is part of any analysis. However, because of STRUCTURE’s complexity, it is harder to conduct the assessment of the model fit. A recently published instructive tutorial [44] provides a critical view on how to assess the outcome of STRUCTURE analysis using the tools badMIXTURE [45], GLOBETROTTER [46], fineSTRUCTURE, and CHROMOPAINTER [47]. In particular, the authors highlighted the fact that different population scenarios (e.g., recent admixture, bottleneck, and admixture contribution from untyped populations) result in similar STRUCTURE barplots [7,48] (the typical data summary from a STRUCTURE analysis). Supplementing the STRUCTURE analysis by residual plots based on badMIXTURE, patterns were detected enabling these scenarios to be distinguished [44]. Similar to PCA, STRUCTURE is sensitive toward biased sampling and population sample sizes. Specifically, imbalances between the analysed populations may influence how and which of the sampled populations that exhibit patterns of population admixture ([44], Case Study 3, pp. 5–8). However, choosing the appropriate (or in a sense, *correct*) value for *K*, the number of population clusters, is not a well-defined problem, i.e., only heuristic methods exist for guiding the specification of *K* [34,35,43,44]. Different choices of *K* may result in rather different results and interpretations of the population stratification and history.

In Figure 2, the estimated STRUCTURE admixture components, $q=(q^{{}_{\left(1\right)}},\cdots ,q^{{}_{\left(K\right)}})$, for the reference samples (cf. above) are shown for K=4. There is a clear visual distinction in the distribution of $q_{i}$ across the 3453 reference samples.

The four clusters of Figure 2, K=4, may be associated with Sub-Saharan African, European, South/Central Asian, and East Asian components. The samples from North African and Middle Eastern regions show the strongest admixture of these four components.

### 2.3. Classification and Likelihood-Based Methods

The statistical problem solved by STRUCTURE [34] and the related methodologies (e.g., ADMIXTURE [41]) is a rather complex one: Assign individuals to populations while estimating the unknown populations’ unknown allele frequencies. As discussed above (Section 2.2), this is performed by finite mixture models, where the assignment of individuals and updating of population allele frequencies are conducted iteratively (either by Bayesian MCMC methods [34] or using efficient variants of expectation–maximisation algorithms [41]). However, if analysis of population structure is not the purpose and the allele frequencies are known for a set of populations, the assignment of individuals is simpler.

In this case, the assignment of nonadmixed individuals can be conducted by a Bayes classifier, where the individual is assigned to the most probable population among the ones included in the reference database. As such, the population assignment problem can be solved by several types of classification algorithms known from the machine learning literature (e.g., classification trees, random forest, support vector machines, multinomial regression, partial least squares, discriminant analysis, nearest neighbours, gradient boosting, and other ensemble learners). Common to most of these methods is their ability to account for interaction effects between the explanatory variables on the outcome (here, the marker interactions when predicting the population of origin). Furthermore, similar to the methods of Section 2.1 and Section 2.2 (e.g., PCA and STRUCTURE), sample sizes and imbalanced training data typically influence the tuning and therefore performance of the machine learning methods. When training a classification algorithm on a specific dataset, the objective is to drive the overall loss function downwards. Typically, the loss function in classification contexts decreases with more samples being accurately classified (e.g., the misclassification rate or smooth functions, thereof). However, if some populations have low proportions in the sample, the classifier gains little from classifying these correct compared to the frequently occurring populations. Hence, the relative composition of the populations in the data may impact the trained classifier quite substantially. In the context of forensic genetics, the risk of incomplete AIMs profiles obtained from, e.g., crime scenes, makes some methods less applicable to samples with some or more untyped markers (e.g., due to low signals or marker drop-out). Some methods (e.g., naïve Bayes classifiers and classification trees using *surrogate splits*) may readily deal with partial profiles, whereas in particular regression, models are susceptible to missing data.

From a forensic perspective evaluating the weight of the evidence is typically conducted using the likelihood ratio principle. For multiple propositions (in this case, multiple populations), this approach generalises to several ratios assessing the pairwise agreement between data and the alleged populations of origin. The predicted population of origin is the population where the profile is most probable. The likelihood function typically assumes a specific population genetic model, e.g., within each population assuming Hardy–Weinberg equilibrium and independent genetic markers. This corresponds to a naïve Bayes classifier assuming a Binomial distribution for each marker given the population specific allele frequencies. Two forensically relevant implementations including several reference populations and AIMs panels are freely available online (FROG-kb (http://frog.med.yale.edu/FrogKB/, accessed on 13 December 2021) [49,50] and Snipper (http://mathgene.usc.es/snipper, accessed on 13 December 2021) [51]). Other approaches toward classification have been considered and compared (e.g., multinomial regression and genetic distances) [52]. These methods are in particular relevant in the case of admixed individuals, where the single population likelihoods do not suffice [53,54].

### 2.4. Hypothesis Test-Based Methods

An underlying and important assumption for both clustering and classification is that of *exhaustive* populations. This means that for clustering the admixture components *must* sum to one, indicating that *all* the genetic admixture of an individual can be explained by contributions from the *K* clusters. However, the *Ghost admixture* example of ([44], p. 3) shows that the noninclusion of important branches of a genealogy may result in misleading results. In that case, the contribution from the unsampled reference population to the admixture was replaced by the reference population most similar to the unsampled one. This scenario may easily occur in the forensic setting, where only major populations are included in the reference material, but where conclusions are wanted on a finer (sub)population scale.

For classifications, an underlying assumption is that an observation belongs to *exactly* one class (i.e., one and only one). Hence, this *forced* classification means that even when all the possible classes are (highly) unlikely, the DNA profile under classification *must* be assigned to precisely one of the populations (typically the *least unlikely* population).

This issue was the main motivation for the derivation of the GenoGeographer methodology [55,56]. Rather than forcing an observation to be classified to any of the prespecified populations in the reference material, it was assessed whether the sample could originate from each of the reference populations. Logically, this implied that the sample was tested for being an outlier in each of the possible populations, and this was formally conducted by testing the hypothesis of the DNA profile sample and database sample originated from the same underlying population or not [55]. This outlier test is equivalent to a Fisher’s exact test and thus enjoys many of the same statistical properties, e.g., increasing power with increasing sample size and robustness toward missing observations. Other hypothesis-based methodologies include a likelihood ratio test for recent admixture [57].

For each reference sample i=1,⋯,3453, STRUCTURE estimates the admixture components, $q_{i}=\left(q_{i}^{{}_{\left(1\right)}},\cdots ,q_{i}^{{}_{\left(K\right)}}\right)$, where $\sum_{k=1}^{K}q_{i}^{{}_{\left(k\right)}}=1$ by definition. From these, the average admixture component is calculated for each of the reference metapopulations, by the average over the admixture components for reference samples belonging to this metapopulation:$${\bar{q}}_{j}=\left({\bar{q}}_{j}^{{}_{\left(1\right)}},\cdots ,{\bar{q}}_{j}^{{}_{\left(K\right)}}\right),\;\mathrm{with}\;{\bar{q}}_{j}^{{}_{\left(k\right)}}=n_{j}^{-1}\sum_{i\in R_{j}}q_{i}^{{}_{\left(k\right)}},\;k=1,\cdots ,K,$$
where $R_{j}$ is the set of the $n_{j}$ samples from metapopulation *j*.

As for the training samples STRUCTURE admixture components $q_{l}$ was calculated for the test samples, l=1,⋯,517. The closer $q_{l}$ is to ${\bar{q}}_{j}$ for some test sample *l* and referece metapopulation *j*, the more likely it is to assume *l* to originate from metapopulation *j*. However, how (dis)similar should these admixture components be to declare *l* (not) to originate from *j*? Moreover, how should this similarity be measured? One possible measure of proximity could be to use a Brier-like score, $B_{lj}$, which is the sum of squared admixture components differences between test sample *l* and reference metapopulation *j*:$$B_{lj}=\frac{1}{K}\sum_{k=1}^{K}{\left(q_{l}^{{}_{\left(k\right)}}-{\bar{q}}_{j}^{{}_{\left(k\right)}}\right)}^{2}.$$

The closer $B_{lj}$ is 0, the more similar is sample *l* to metapopulation *j* in terms of STRUCTURE admixture components (Supplementary materials of [26]). The maximal value of $B_{lj}$ is 2/K, which happens if $q_{l}^{{}_{\left(k\right)}}=1$ and ${\bar{q}}_{j}^{{}_{\left(k^{\prime }\right)}}=1$ for $k\neq k^{\prime }$. In Figure 3, the boxplots of the Brier scores for each of six metapopulations are grouped according to the test samples’ status inferred by the GenoGeographer methodology (in colours) [55]. Added to plot in grey boxplots are the Brier scores for the reference samples, which are the samples used to compute ${\bar{q}}_{j}$. Thus, the grey boxplots are expected to show lower Brier scores compared to those of the coloured boxplots.

The Brier scores, $B_{ij}$, in Figure 3, tend to be larger for discordant, ambiguous, and rejected samples (see definitions in caption of Figure 3). However, from the visual inspection of the STRUCTURE barplots (data not shown), it is often rather hard to judge when a sample deviates substantially from a given metapopulation to declare it extreme compared to the other samples. The advantage of GenoGeographer is that such visual inspections or *ad hoc* defined thresholds for dissimilarity (e.g., using a Brier score) are not required. The conclusions are based solely on a hypothesis test and a predefined significance level.

The model formulation of STRUCTURE implies that any spurious and complex type of admixture can be modelled. For the GenoGeographer framework, however, only specific admixtures can be assessed. Currently, first-order admixtures can be accounted for [56], but the admixture approach can be generalised to handle outlier tests for higher-order admixtures (e.g., second-order where the parents themselves may be admixed). However, the ancestry SNPs may not be sufficiently informative to distinguish between certain admixture configurations (e.g., two pedigrees with different founder populations may be equally likely).

One solution to this may be to use linked markers as these typically are more informative for pedigree analyses [59]. In order to handle linked markers in the framework of hypothesis tests, e.g., microhaplotypes [17], the interactions between the markers was modelled using decomposable graphical models, where association structures between the SNPs could be different among the reference populations [23] (e.g., the linkage between SNPs was allowed to be different between continental regions). For a microhaplotype with *K* biallelic SNPs, there exist $2^{K}$ different alleles, implying that many samples are needed to obtain accurate allele frequency estimates for moderate values of *K*. By exploiting the conditional independence structures, the framework of graphical models provides a more data efficient modelling of the microhaplotype frequencies and thus requires fewer samples.

## 3. Discussion

STRUCTURE remains to be a valuable methodology and approach for analysing population structure in forensic genetics. There exist several guides and tutorials for how to prepare the population data and choosing parameter settings for STRUCTURE (e.g., [60,61], with some emphasis on forensic applications). However, as warned by [44], the resulting barplots should not be overinterpreted. The badMIXTURE approach [45] provides the means to reduce the risk of being mislead by the summaries provided by STRUCTURE and supporting software (e.g., [7,48]).

However, STRUCTURE’s popularity in the field of population genetics to study population structure is well deserved. It complemented PCA with a quantitative method that assigns sample specific weights to each of the *K* populations specified in the study. This is a powerful way to summarise the data in terms of cluster membership probabilities and the estimated allele frequencies for the identified populations. Both PCA and STRUCTURE are valuable for exploratory analysis, where encoding errors (e.g., of missing data) and warnings about misspecification of origin may be detected by visual inspections of the results. From a forensic point of view, the results from both PCA and STRUCTURE are hard to report in terms of a weight of evidence calculation. The similarity (or dissimilarity) between the sample and reference materials can be reported but typically only in terms of their visual appearance. By contrast, classification and likelihood-based methods are able to give weight to the evidence. This may be in terms of a posterior probability or likelihood ratio, where the assignment would be to the most probable population. However, none of these methods take into account the risk of assigning the profile to the *least unlikely* population in the case where none of the populations are representative for the true population of origin.

Inferring a DNA profile’s geographic region or population of origin has been an active field of research in forensic genetics for several years. The vast collections of publicly available databases of whole-genome sequences provide a unique resource for researchers. Rather than spending time and consumables on collecting samples, these DNA profiles can now be harvested online. However, in doing so, one relies on the quality of the data provided by others, which includes the genetic typing and base calling but also the metadata regarding sampling location and ethnic information.

In the case of forensic genetics, the finer resolutions are often of interest. In order to increase the specificity, it may be necessary to supplement the selected AIMs with more locally informative markers, specific to separating the local and often related populations of interest. In the search for such markers, the publicly available datasets are of immense importance as they can be used to screen for candidate AIMs. In particular, the samples from regions close to the specific populations of interest are essential for selecting potential markers.

Combining the flexibility of STRUCTURE with the appealing visual features of PCA (e.g., using EIGENSTRAT to determine the number of significant components) is essential in the exploratory phase of ancestry inference. However, the forensic questions related to ancestry is typically different from those of population genetics, where the focus may be on *population* specific patterns (e.g., bottleneck and expansion events or migration). The typical forensic use case focuses on a specific *individual* (or human remains), for which the most likely population of origin needs to be identified. In such cases, likelihood ratio or classification methods can be used. However, the fundamental assumption of the existence of an appropriate population in the reference material may very often be violated. Hence, DNA profiles are assigned to the population that is most similar to the sample, which due to human evolution and history may geographically and culturally be very far away from the true population. The GenoGeographer methodology attempts to overcomes this logical problem by using statistically based outlier tests.

## Figures and Tables

**Figure 1 genes-13-00141-f001:**
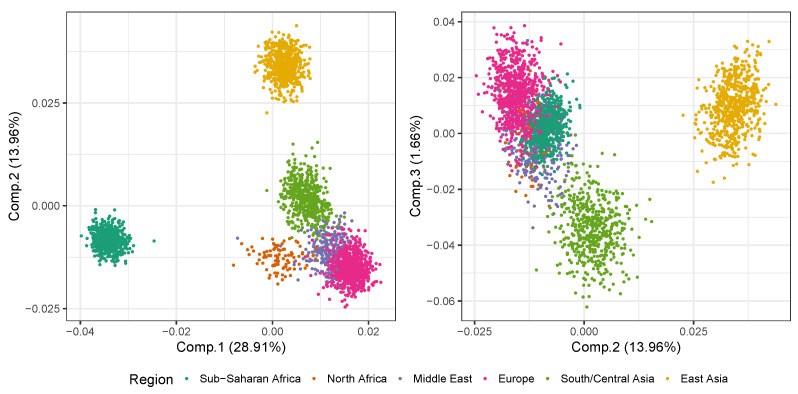
Plot of the first three PCs from the PCA of the reference samples.

**Figure 2 genes-13-00141-f002:**
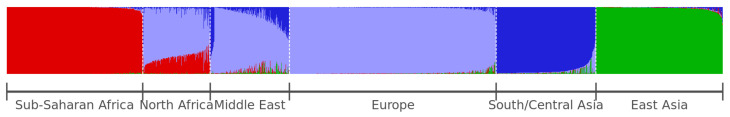
Barplots of the STRUCTURE admixture components (K=4) for the reference samples from six metapopulations. There is a clear visual difference between the admixture components for the six metapopulations.

**Figure 3 genes-13-00141-f003:**
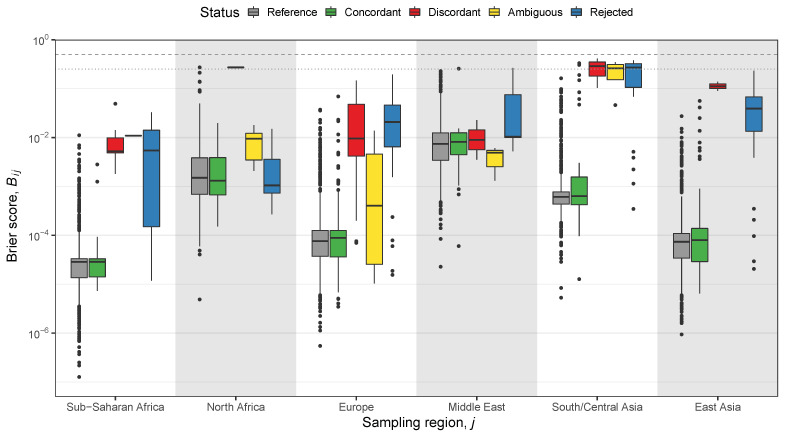
Boxplots of the Brier scores, $B_{ij}$, with K=4 in the STRUCTURE analysis and *j* running through the metapopulations of [58]. *Reference:* Samples constituting the metapopulations (and used to calculate ${\bar{q}}_{j}$). *Concordant:* A sample is accepted in the metapopulation coinciding with its sampling region (with a likelihood value significantly larger than any other metapopulation). *Discordant:* A sample is accepted in a metapopulation different than its sampling region (with a likelihood value significantly larger than any other metapopulation). *Ambiguous:* A sample’s likelihood value is not significantly different in two or more of the accepted metapopulations. *Rejected:* A sample is rejected in all metapopulations. The concordant test samples (green) have similar Brier scores as those of the reference samples (grey). Discordant samples (red) tend to have the largest Brier scores, followed by the rejected samples (blue) with the ambiguous (yellow) in between (see text for definitions). The horizontal dashed and dotted lines in the top of the plot indicates $B_{ij}=0.5$ and 0.25, respectively ([26], inspired by Supplementary Figure S8).

## Data Availability

Not applicable.

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
