# Peer review of "Review of the Forensic Applicability of Biostatistical Methods for Inferring Ancestry from Autosomal Genetic Markers"

_genes, 2022, doi:10.3390/genes13010141_

Round 1

Reviewer 1 Report

This is a review of the manuscript “Review of Biostatistical Methods for Inferring Ancestry from Genetic Markers” that was submitted to Genes (genes-1532554) by Torben Tvedebrink.

In this manuscript the author focusses on the use of autosomal DNA polymorphisms for the inference of the ancestry of an individual. As such, it specifically excludes the use of Y-chromosome and mitochondrial DNA based ancestry inference methods. Hence, I would recommend to insert the word “autosomal” between “from” and “genetic” in the title.

It is otherwise an excellent review of the available methods, with a well-balanced discussion of the various pro’s and con’s of each of the methods discussed. However, at one point I find the manuscript difficult to understand and follow. This relates to the lines 212-234. In this part, the author directly compares the use of STRUCTURE with GenoGeographer. For this he uses the so-called Brier score. This section needs much more explanation and more graphical support. E.g. it is unclear what part of this section relates to actual empirical data or is simulated.

Furthermore, I think it would be helpful to include, into the section lines 102 – 138 two graphical representations showing the difference between PCA and STRUCTURE.

And finally, at several places in the manuscript the author refers to “loss function” without further explanation. Perhaps this could be remedied.

Author Response

This is a review of the manuscript “Review of Biostatistical Methods for Inferring Ancestry from Genetic Markers” that was submitted to Genes (genes-1532554) by Torben Tvedebrink.

In this manuscript the author focusses on the use of autosomal DNA polymorphisms for the inference of the ancestry of an individual. As such, it specifically excludes the use of Y-chromosome and mitochondrial DNA based ancestry inference methods. Hence, I would recommend to insert the word “autosomal” between “from” and “genetic” in the title.

I welcome the reviewer’s suggestion and have added “autosomal” to the title for better agreement with the context of the paper. Furthermore, Reviewer 2 also comments on the title, which led to the following change to the title: “Review of the Forensic Applicability of Biostatistical Methods for Inferring Ancestry from Autosomal Genetic Markers”.

It is otherwise an excellent review of the available methods, with a well-balanced discussion of the various pro’s and con’s of each of the methods discussed. However, at one point I find the manuscript difficult to understand and follow. This relates to the lines 212-234. In this part, the author directly compares the use of STRUCTURE with GenoGeographer. For this he uses the so-called Brier score. This section needs much more explanation and more graphical support. E.g. it is unclear what part of this section relates to actual empirical data or is simulated.

I have attempted to clarify that real data has been used in the analysis and comparison of STRUCTURE and GenoGeographer. The section has been expanded to allow for (hopefully) a clearer explanation of the introduced concepts.

Furthermore, I think it would be helpful to include, into the section lines 102 – 138 two graphical representations showing the difference between PCA and STRUCTURE.

I agree and I have added a plots trying to compare the two methods and highlight their differences.

And finally, at several places in the manuscript the author refers to “loss function” without further explanation. Perhaps this could be remedied.

The use of “loss function” has been clarified and its usage been made more concrete.

Reviewer 2 Report

The author synthesises available methodologies for modelling population admixture and inferring population ancestry. Focus is given to the strengths and weaknesses in ancestry inference in the human forensic context. The structure of the manuscript is deliberate, its language is good, however, based on the title, I thought the statistical background of the methods would be more emphasized. The title should be reconsidered.

However, I found the manuscript enjoyable and informative. But I miss the mentioning of the DAPC method:

Jombart et al. Discriminant analysis of principal components: a new method for the analysis of genetically structured populations. BMC Genetics 2010, 11, 94 https://doi.org/10.1186/1471-2156-11-94

Miller et al. The influence of a priori grouping on inference of genetic clusters: simulation study and literature review of the DAPC method. Heredity 2020, 125, 269-280.  https://doi.org/10.1038/s41437-020-0348-2

Minor comments

Lines 76-77: It is unnecessary to explicitly denote the subsequent sections in “In the subsequent sections (Sections 2.1 – 2.4)…”

Line 106: Abbreviations, as MCMC in this case, should be defined at first mention.

Line 145: Abbreviations, as EM in this case, should be defined at first mention

Author Response

The author synthesises available methodologies for modelling population admixture and inferring population ancestry. Focus is given to the strengths and weaknesses in ancestry inference in the human forensic context. The structure of the manuscript is deliberate, its language is good, however, based on the title, I thought the statistical background of the methods would be more emphasized. The title should be reconsidered.

In line with Reviewer 1’s comment on the title, the title of the paper has been adjusted to better reflect the content of the paper. Now the title reads “Review of the Forensic Applicability of Biostatistical Methods for Inferring Ancestry from Autosomal Genetic Markers”

However, I found the manuscript enjoyable and informative. But I miss the mentioning of the DAPC method:

Jombart et al. Discriminant analysis of principal components: a new method for the analysis of genetically structured populations. BMC Genetics 2010, 11, 94 https://doi.org/10.1186/1471-2156-11-94

Miller et al. The influence of a priori grouping on inference of genetic clusters: simulation study and literature review of the DAPC method. Heredity 2020, 125, 269-280.  https://doi.org/10.1038/s41437-020-0348-2

Thank you for adding to the list of methodologies. I’m sorry to have missed the DAPC-methodology in my original literature review. I have added the two references to the manuscript in Section 2.1 Principal components analysis (lines 90 – 95).

Minor comments

Lines 76-77: It is unnecessary to explicitly denote the subsequent sections in “In the subsequent sections (Sections 2.1 – 2.4)…”

Deleted

Line 106: Abbreviations, as MCMC in this case, should be defined at first mention.

I have added Markov Chain Monte Carlo in the first appearance of MCMC

Line 145: Abbreviations, as EM in this case, should be defined at first mention

Amended.